

# Temporal and spatial refugia modify predation risk for non-native crabs in rocky intertidal habitats

Renee C. Montanaro and Nancy J. O'Connor

Biology Department, University of Massachusetts at Dartmouth, Dartmouth, MA, United States of America

## ABSTRACT

Populations of the non-native Asian shore crab, *Hemigrapsus sanguineus*, now dominate the rocky intertidal of southern New England, USA. High abundances suggest the recent invader may have experienced enhanced success as a result of enemy release. While larvae and juveniles may serve as a food source for ecologically important species, little is known about predation of mature *H. sanguineus* or the influence of habitat on predation pressure. To assess natural predation rates of adult *H. sanguineus,* crabs were tethered in the intertidal at Clarks Cove in New Bedford, MA. Crabs were left *in situ* for half of a tidal cycle then observed for signs of predation. Results of separate high and low tide trials showed that adult crabs were preyed upon at both high and low tide, though at a significantly higher rate during high tide during both daytime and nighttime, suggesting predation by aquatic species is greater than that by terrestrial species. To investigate the role of habitat as refuge from predation, a laboratory experiment manipulated the complexity of habitat provided to crabs in the presence of a native fish predator. Results indicate better refuge is provided by more complex shelter. Together, findings suggest that fish, crabs, and/or diving birds are important predators for *H. sanguineus* in the invaded range and that habitat refuge acts to reduce predation pressure.

## INTRODUCTION

A prominent factor thought to facilitate the proliferation of non-native populations is enemy release (*Heger & Jeschke, 2018*), wherein non-native species benefit from a reduction in natural predators, competitors, and parasites in the naïve systems they invade (*Colautti et al., 2004*). While enemy release may confer an advantage to non-native species (*Antonini et al., 2019*; *Roznik et al., 2020*), invasion success is dependent on a myriad of ecological and environmental interactions within native communities (*Weis, 2010*; *Prior et al., 2015*).

Across taxa and environments, the availability of temporal and spatial refuge can act to constrain predation risk (*Prugh & Golden, 2014*; *Palmer et al., 2022*; *Suraci et al., 2022*). Habitat structure, the arrangement of biotic and abiotic substrate that supports plant and animal communities (*Carvalho & Barros, 2017*), can mediate interactions between predators and prey by providing spatial refuge in which prey can more easily avoid capture (*Warfe & Barmuta, 2004*; *Lei, Lin & Zhang, 2014*; *Mendez, Schwindt & Bortolus,*

Corresponding author
Renee C. Montanaro,
renee.montanaro@umassd.edu

*2015*; *Pozzebon, Loeb & Duso, 2015*). In addition to habitat-specific structural refuge, predation risk can change over time (*Sperry et al., 2008*). For example, risk and refuge for prey can exhibit diel variation (*Clark, Ruiz & Hines, 2003*). More to the point, the availability of refuge is a consequential determinant of predation risk (*Smith et al., 2019*), but rarely explored as a factor influencing enemy release (*Soifer & Ackerman, 2019*).

Evidence suggests a recent invader to the rocky shores of the North American Atlantic coast, the Asian shore crab, *Hemigrapsus sanguineus*, may have experienced enhanced success as a result of enemy release. The abundance of *H. sanguineus* in invaded habitats can far exceed densities found along the native Asian-Pacific region (*Takahashi et al., 1985*; *Lohrer et al., 2000*). Today, *H. sanguineus* is the most abundant intertidal crab species in southern New England and Long Island Sound (*Kraemer et al., 2007*; *O'Connor, 2014*) and can reach densities of >300 crabs m$^{-2}$ at some locations (*O'Connor, 2018*). The species' success since its introduction to the northeast US in the 1980s is thought to be explained, in part, by reduced impact of natural enemies compared to the populations in native habitats (*Pushchina & Panchenko, 2002*; *Brousseau et al., 2008*).

The relationship between the non-native *H. sanguineus* and native predators, however, remains ambiguous in the absence of direct field experimentation and laboratory tests with adult crabs. In its native range, *H. sanguineus* is known to be consumed by two species of sculpins, *Myoxocephalus stelleri* and *M. brandti* (*Pushchina & Panchenko, 2002*). Species thought to prey on non-native *H. sanguineus* include those that utilize the rocky intertidal zone to forage, and species that are adapted to eating hard-shelled benthic invertebrate prey, including fish and likely bird species (*Epifanio, 2013*).

Of the potential predators of non-native *H. sanguineus* present throughout its range in northeast North America, only predation by fish on larvae and juveniles has been examined (*Kim & O'Connor, 2007*; *Rasch & O'Connor, 2012*). Little is known about predation on larger, sexually mature *H. sanguineus.* In one laboratory choice experiment, Tautog (*Tautoga onitis)* consumed juvenile *H. sanguineus*, but less often than native prey species (*Savaria & O'Connor, 2013*). Conversely, other laboratory experiments found *T. onitis*, cunner (*Tautoga adspersus)*, and black sea bass (*Centropristis striata*) preferentially preyed upon *H. sanguineus* when given the choice with other local crab species as prey and that substrate influenced predator preference (*Heinonen & Auster, 2012*). Additionally, factors that modify predator–prey relationships, like spatial and temporal refuge, should be more fully explored to clarify the factors influencing enemy release in invaded systems.

The purpose of this study was to measure predation of sexually mature *H. sanguineus* in the field to examine temporal refuge from predation risk and use laboratory experiments to assess the influence of spatial refuge on predation. We hypothesized that relative predation of adult *H. sanguineus* is greatest during daytime high tides, when visual fish predators are active, and that predation risk by the fish predator *T. onitis* is mitigated by high levels of habitat refuge.

## MATERIALS & METHODS

### Field experiment

#### Crab collection and housing

Male ($n = 28$) and female ($n = 6$) sexually mature *H. sanguineus* (15–22 mm in carapace width, CW) used for tethering experiments were collected at the study site in Clark's Cove, New Bedford, Massachusetts (41°35′40.33″N, 70°54′37.45″) by hand at low tide and outfitted with tethers 12–24 h prior to experimentation. Crabs were held individually in 113 L aerated aquaria, with water sourced directly from Clark's Cove, and kept at ambient conditions (temp 20.5–24.5 °C, salinity 33–34), housed adjacent to the field site at the University of Massachusetts School for Marine Science and Technology Seawater Lab. Outfitting crabs with tethers beforehand ensured that the tether was retained and did not impede mobility.

The tethering apparatus was constructed using 0.3 m of monofilament fishing line (6.8 kg test strength) secured to the crab by looping around the transverse plane of the body between the 2nd and 3rd walking legs. The line was tied at the dorsal midline and the knot was secured with a drop of cyanoacrylate glue. Crabs were reliably recovered using this tethering method and the tether was shown not to cause damage to the crabs when subjected to simulated wave energy in the lab.

#### Field tethering procedure

Tethering is a useful method to compare relative predation intensity (*Moody & Aronson, 2007*; *Glazner, Ballard & Armitage, 2021*). Field tethering experiments were conducted in the lower intertidal zone of Clark's Cove, New Bedford, MA (41°35′40.33″N, 70°54′37.45″W) June 28–August 26, 2020 (Table 1) at +0.29 m above mean low water (average tidal range for the location is 1.1 m). Previous investigations at the study site showed crabs were most abundant at that tidal elevation (*Towne, Judge & O'Connor, 2023*).

To prepare the field site for tethering experiments, all rocks and cobble were removed from a 1 m diameter circle, so that only flat sandy substrate was available to the tethered crab. At the time of experimentation, the free end of the tether line was attached to the top of a 10 cm stake embedded in the sediment. Two experimental replicates (plot A and B) were established >10 m apart. The experiments took place during low tide and high tide, during daytime (daylight hours) and nighttime (after sunset). Tethered crabs were left *in situ* for half of a tidal cycle (beginning three hours before low/high tide and ending three hours after low/high tide). Crabs were considered to have been eaten by predators if missing at the end of the trial. Each crab was used for a single trial, and surviving crabs were returned to the wild, outside of the area where this work was performed.

#### Field tethering analysis

Even some of the low tide trials were wet during a portion of the experiment because of the position low in the intertidal, and the sun rose or set during some of the trials, therefore, analysis was performed using the minutes a crab was submerged and the minutes of daylight during a trial. To test whether tide (minutes submerged) and time (minutes

**Table 1  Details of field experiment conditions.** Dates experimental trials were conducted in 2020, start (time the trial started), end (time the trial ended), sun rise (time the sun rose) sun set (time the sun set), minutes of daylight (amount of daylight during each trial), predicted tidal height (difference from mean low water level in meters, source: US Harbors Padanaram, South Dartmouth, MA, USA; https://www.usharbors.com/harbor/massachusetts/padanaram-south-dartmouth-ma/), tide time (time of the peak high/low tide), and the amount of time crabs were submerged (minutes submerged).

| Trial # | Date | Start | End | Sun rise | Sun set | Minutes of daylight | Tidal height (m) | Tide time | Minutes submerged |
|---|---|---|---|---|---|---|---|---|---|
| 1 | 28-Jul | 5:35 | 11:35 | 5:35 | 20:05 | 360 | 0.1 | 8:30 | 120 |
| 2 | 28-Jul | 19:10 | 1:00 | 5:35 | 20:05 | 65 | 0.2 | 22:00 | 150 |
| 3 | 29-Jul | 13:15 | 19:30 | 5:35 | 20:00 | 375 | 1.3 | 16:30 | 375 |
| 4 | 30-Jul | 2:10 | 8:25 | 5:35 | 20:00 | 195 | 1 | 5:00 | 375 |
| 5 | 3-Aug | 5:35 | 11:30 | 5:40 | 20:00 | 355 | 1.1 | 8:25 | 355 |
| 6 | 4-Aug | 6:00 | 12:00 | 5:40 | 20:00 | 360 | 1.1 | 9:10 | 360 |
| 7 | 12-Aug | 12:15 | 18:20 | 5:50 | 19:45 | 365 | 1 | 17:25 | 365 |
| 8 | 13-Aug | 5:45 | 11:50 | 5:50 | 19:45 | 365 | 0.2 | 8:50 | 200 |
| 9 | 13-Aug | 7:00 | 13:05 | 5:50 | 19:45 | 365 | 0.2 | 9:50 | 180 |
| 10 | 18-Aug | 10:40 | 16:50 | 5:55 | 19:40 | 370 | −0.1 | 13:40 | 110 |
| 11 | 19-Aug | 11:30 | 17:30 | 5:55 | 19:35 | 360 | −0.1 | 14:25 | 100 |
| 12 | 19-Aug | 23:35 | 6:00 | 6:00 | 19:35 | 240 | −0.1 | 2:55 | 125 |
| 13 | 20-Aug | 13:00 | 19:00 | 6:00 | 19:35 | 360 | −0.2 | 15:15 | 135 |
| 14 | 21-Aug | 0:30 | 6:40 | 6:00 | 19:30 | 100 | −0.2 | 3:40 | 145 |
| 15 | 24-Aug | 10:00 | 15:50 | 6:00 | 19:30 | 350 | 1.3 | 13:05 | 350 |
| 16 | 25-Aug | 11:00 | 16:30 | 6:00 | 19:30 | 330 | 1.3 | 14:00 | 330 |
| 17 | 26-Aug | 12:00 | 18:00 | 6:05 | 19:25 | 360 | 1.2 | 15:10 | 360 |

of daylight) influence predation of adult crabs, predation was examined using binomial regression. The dependent variable (predation) was coded as binary data (predation = 1; no predation = 0). The test determined the probability that a crab would be eaten based on the independent variables tide (minutes submerged), and time (minutes of daylight). Significance of factors was evaluated with analysis of deviance using the anova() function of the car v3.1-2 package in R (*Fox & Weisberg, 2019*). All statistical analyses performed in this study were done using R v4.0.0 (*R Core Team, 2020*).

## Laboratory experiment
### Crab and fish collection

The tautog (*T. onitis*) is a temperate reef fish that plays an important role in the structure of nearshore marine communities as a specialized predator of hard-shelled benthic invertebrates, including crabs (*Liem & Sanders, 1986*; *Clark et al., 2006*). The 27 *T. onitis* (25.5–37 cm total length) used in the laboratory experiment were caught in New Bedford Harbor, MA using unbaited traps May 13–June 4, 2021. Fish were collected during annual trap surveys conducted by the Massachusetts Division of Marine Fisheries; traps were checked at least every three days. Fish were transported in a 50 L insulated cooler with fresh seawater and continuous aeration. Fish were held in groups of <10 for a two-week acclimation period ahead of experimentation. During acclimation, fish were fed crabs and cracked clams to satiation. Fish and crabs were housed, and experiments were conducted,

in 1.8 m diameter tanks (tank floor area = 2.6 m$^2$) continuously supplied with ambient seawater from Clarks Cove, New Bedford (water temp 20.5–24.5 °C, salinity 33–34, depth 1 m) and artificially lit to match natural light-dark cycles. Tanks were cleaned daily. Fish were provided pieces of large PVC pipe (10.2 cm diameter) for shelter during acclimation. Each fish was used for a single trial, then returned to the wild.

The 720 *H. sanguineus* (14–20 mm CW) used in these tests were collected from the rocky intertidal in Clark's Cove 12–24 h prior to experimentation. Crabs were held in one tank in mesh-sided 0.5 L Tupperware containers in groups of <6 crabs. Crabs were not fed during this time. Only non-gravid crabs with all ten limbs were used in this study. Each crab was used for a single experiment.

### Lab experiment treatment construction

Habitat structure provided during the lab experiment was constructed from concrete pavers (paving stones) (*L* = 40 cm, *W* = 20 cm, *H* = 5 cm). The experiment included a No Refuge control treatment without structure, as well as a control treatment without a fish predator. Low Refuge Habitat consisted of two pavers laid flat on the bottom of the tank (Fig. 1). High Refuge Habitat consisted of two pavers, modified to create sloping structure by the addition of twelve quartzite river stones (2–3 cm) glued to one long edge of the pavers using saltwater resistant Seachem cyanoacrylate Reef Glue™. This created 1,984 cm$^3$ of refuge space under each paver (Fig. 1). All materials were rinsed with fresh water and allowed >24 h to air dry before use. The glue was given >24 h to cure. Stones remained glued in place throughout the duration of the experiment.

Each tank was outfitted with a shelter for the fish made of three large PVC tubes (10.2 cm diameter), suspended eight cm above the tank floor in the center of the tank (Fig. 1). Fish utilized the inside and the outside of the PVC tubes as shelter. These PVC tubes were not accessible to the crabs. Fish were maintained and housed under University of Massachusetts Dartmouth Institutional Animal Care and Use Committee protocol # 21-02 approved July 19, 2021.

### Lab experimental procedure

Four experimental tanks were randomly assigned habitat treatments for each trial. Nine trials were conducted for each habitat treatment (High Refuge, Low Refuge, No Refuge, and no fish control). One fish was used per trial and allowed 24 h to acclimate in the experimental tank prior to the experiment, during which time the fish was not fed. *T. onitis* require 8 h to process and evacuate food (*Olla, Bejda & Martin, 1975*). To begin the experiment, habitat treatments were lowered into the tanks (pavers placed >0.5 m away from each other and from tank walls) and 20 crabs were added to the tank. A dip net was used to quarantine the fish for 15 min to allow the crabs to acclimate to the experimental tank. Fish were then given 6 h to feed. Results of pilot work showed, when starved for 24 h, one *T. onitis* (35 cm total length) could consume between 20 and 30 *H. sanguineus* (10 mm–18 mm CW) in 6 h. All trials were conducted during daylight hours, approximately 0700–1300.

At the end of each trial, the surviving crabs were counted. Fish were returned to storage tanks and observed for an hour after experimentation, then transported in a cooler with

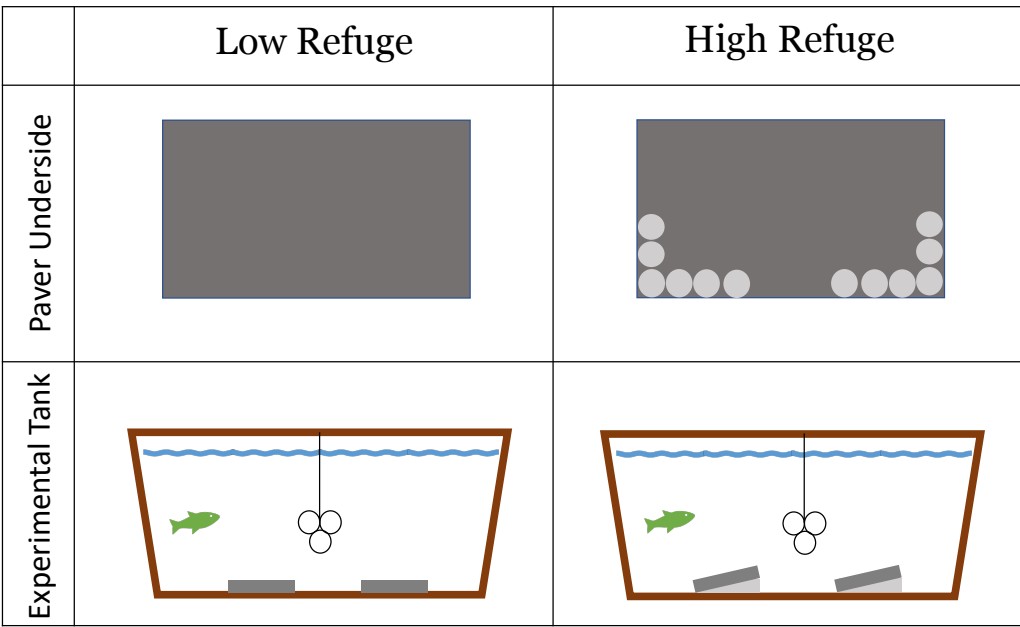

**Figure 1 Low refuge habitat and high refuge Habitat treatments in the laboratory experiment with concrete pavers.** Paver underside: dark gray area shows the underside of the 20 × 40 cm paver used to construct habitat; light gray circles represent individual quartzite river stones (2–3 cm) glued to the underside of the paver on one side. Experimental tank profile: dark gray represents habitat pavers and light gray represents space available to crabs underneath pavers. The line and three circles in the middle of the experimental tank show the shelter that was available to the fish.

sea water to Clark's Cove. No fish showed signs of illness or injury, and all 27 fish used in experiments were released. Tanks were drained and cleaned following each trial.

### *Lab experimental analysis*

The proportion of crabs eaten was calculated for all trials. To test whether habitat refuge treatments influenced the proportion of crabs eaten, all habitat treatments were compared using a one-way ANOVA. A Tukey *post hoc* test was then performed to determine any differences among levels of habitat complexity. Other independent variables including fish size, water temperature, and experimental tank (A, B, C, D) were tested using an expanded ANOVA model, none of which had significant influence on the proportion of crabs eaten, so were excluded from the final analysis. Data met the assumptions of normality and homogeneity of variance. Significance of factors was evaluated with type-III sums of squares using the R package car (*Fox & Weisberg, 2019*).

## RESULTS

### Field tethering

One male and one female crab were found damaged (lost multiple limbs) after experiments, and although injuries were likely the result of predation attempts by a small predator, injured crabs were excluded from the analysis. Of the remaining 32 crabs, 17 were missing and presumed eaten, and 15 crabs were recovered unharmed. There was a significantly

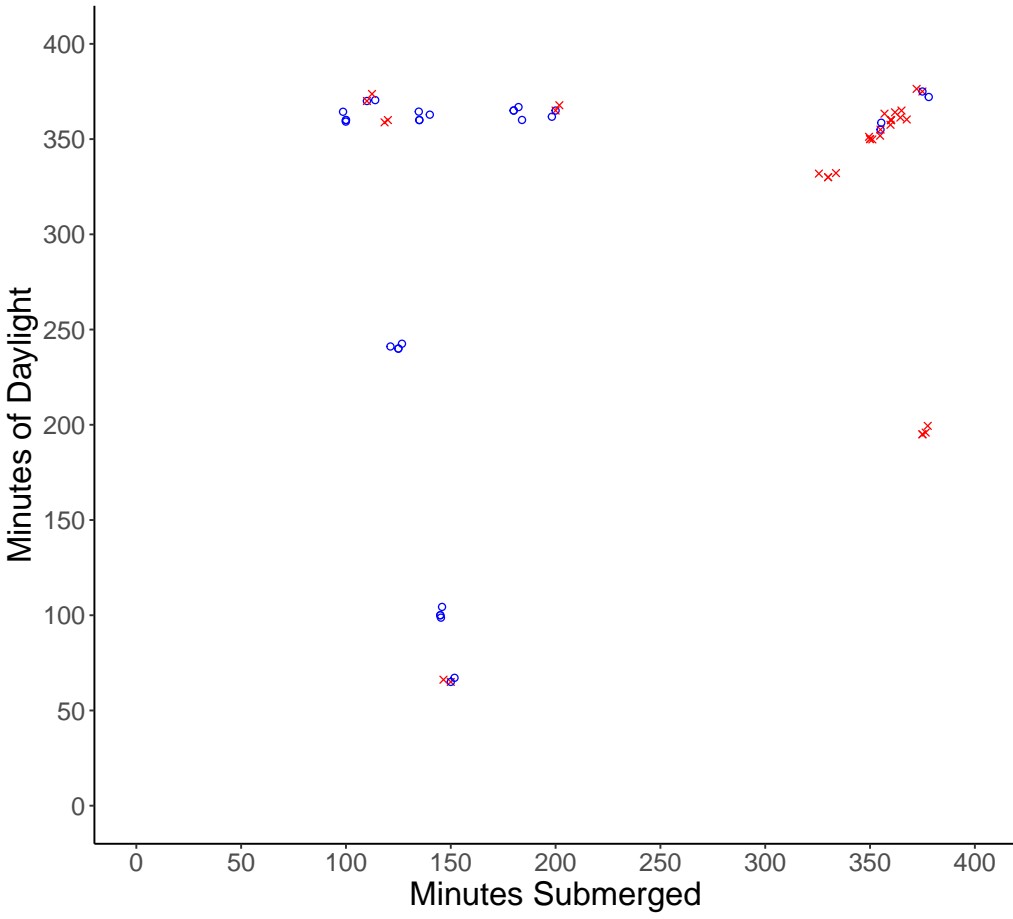

**Figure 2   Results of tethering experiments in the field.** Experiments in which predation occurred are represented by red 'x' symbols, experiments in which no predation occurred are represented by blue circles.

higher probability of predation with increased time of submersion ($df = 30$, $X^2 = 30.997$, $p < 0.001$), while the amount of daylight did not influence the probability of predation ($df = 29$, $X^2 = 30.987$, $p = 0.92$) (Fig. 2).

## Laboratory experiment

In the control (no fish) treatment, all crabs survived without injury. Predation in laboratory feeding trials varied significantly with habitat complexity ($df = 2$, $F = 35.99$, $p < 0.001$) (Fig. 3). The proportion of crabs eaten was significantly lower in the presence of High Refuge Habitat compared to both Low Refuge Habitat ($p < 0.001$) and No Refuge ($p < 0.001$). There was no difference in predation between the Low Refuge Habitat and No Refuge treatments ($p = 0.63$) (Fig. 3). One fish consumed all 20 crabs available in the experiment (No Refuge, fish total length = 31 cm), and one fish consumed no crabs (High Refuge Habitat, fish total length = 35.5 cm).

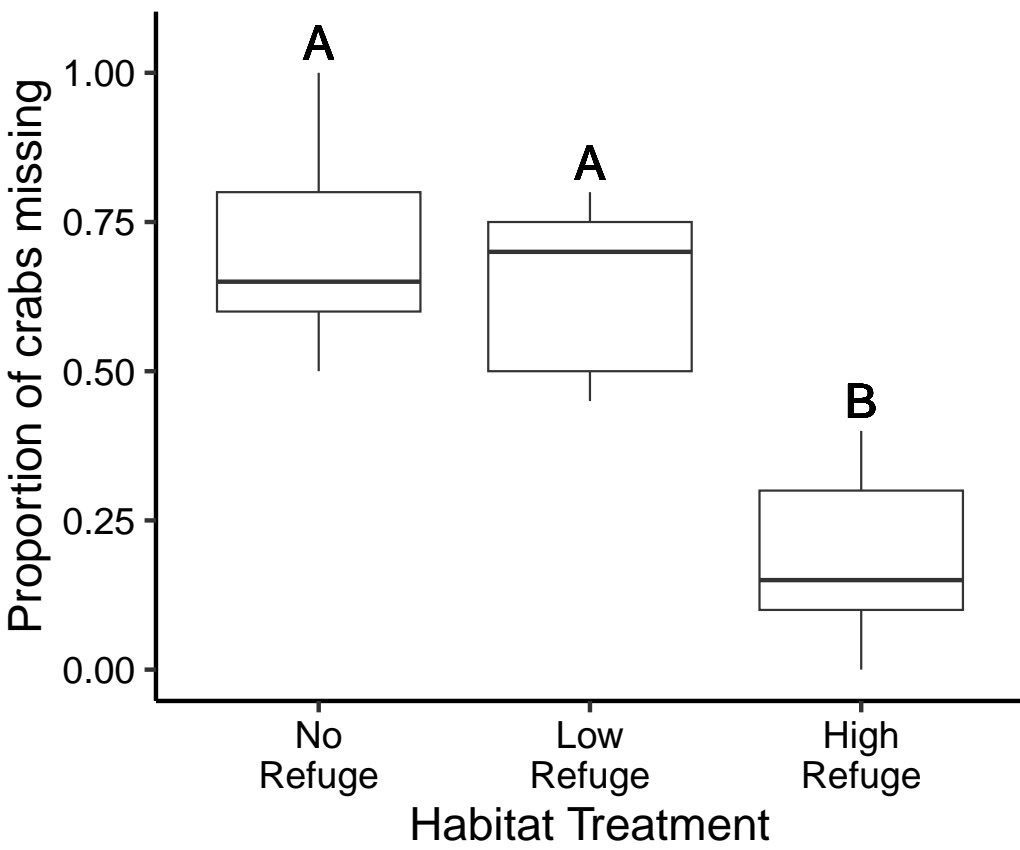

**Figure 3** **Results of habitat complexity experiments in the laboratory.** Box and whisker plot showing the proportion of crabs missing with No Refuge, Low Refuge Habitat and High Refuge Habitat. Boxes indicate 25th percentile (Q1), median, and 75th percentile (Q3). Whisker lines extend to maximum and minimum values. Habitat treatments that share a letter were not significantly different (Tukey $p < 0.001$). Sample size was equal for all habitat treatments ($n = 9$).

## DISCUSSION

This study investigated temporal refuge by explicitly removing the spatial refuge, and spatial refuge by eliminating temporal refuge. Without spatial refuge predation of *H. sanguineus* was influenced by tide and not affected by time of day in field experiments. Crabs had a higher probability of being eaten when they were submerged in water. However, predation rates were similar during the day and night both at high tide and low tide. The tethering experiment in the field likely enhanced actual predation risk, because the tether apparatus restricted the crabs' ability to flee predators, and the experiment was conducted in the absence of refuge for the crab. Nevertheless, results show predation of adult *H. sanguineus* can occur during high tide and low tide, and during daytime and nighttime, suggesting that diurnal as well as nocturnal marine and terrestrial species can prey on the non-native *H. sanguineus*.

The fish species *T. onitis* and *T. adspersus* were likely the primary fish species responsible for predation at high tide during this study. The native fishes are specialized for consuming

hard-shelled invertebrates in the intertidal zone during high tide (*Olla et al., 1974*), and are known to feed on crabs, including *H. sanguineus* (*Clark et al., 2006*). *T. onitis* is a member of the Labridae family, named for their terminal protractile mouths and jaw morphology associated with hard-prey diets (*Liem & Sanders, 1986*), and feeds predominantly on mussels and other shellfish, including Atlantic rock crabs (*Cancer irroratus*), Jonah crabs (*Cancer borealis*) and small American lobsters (*Homarus americanus*) (*Smith, 1907*; *Steimle & Ogren, 1982*; *Richards, 1992*). At the study site, in >20 h of video footage of tethered crabs recorded during pilot work (July 18–August 23, 2020) there were three predation attempts observed during daytime high tides and all appeared to involve *T. onitis* and/or *T. adspersus* (R. Montanaro, pers. obs., 2020 ).

Higher probability of predation during high tide may, in part, be explained by what is known about the behavior of *H. sanguineus*. Both male and female *H. sanguineus* are highly mobile and can travel up to 16 m per day (*Brousseau et al., 2002*). Thought to be most active at high tide, laboratory experiments conducted during daytime found crabs were more likely to move under and out from under shelter when submerged in water (*Towne, Judge & O'Connor, 2023*). Activity at high tide may expose crabs to visual predators, like *T. onitis* and *T. adspersus*, which use a scan-and-pick foraging strategy to feed in the intertidal zone (*Dew, 1976*; *Deacutis, 1982*) where the crabs occur (*Brousseau et al., 2002*; *Epifanio, 2013*; *Towne, Judge & O'Connor, 2023*).

Given greater relative predation at high tide, periods of low tide appear to offer, to some extent, temporal refuge from predation. While at lower levels, predation did occur during field tethering experiments at low tide, demonstrating that species that are not strictly aquatic also pose a predation risk to adult *H. sanguineus*. Other invertebrate species could have been responsible for the predation observed in this study at high tide or low tide. European green crabs, *Carcinus maenas*, Atlantic rock crabs, *C. irroratus*, and blue crabs, *Callinectes sapidus*, co-occur with *H. sanguineus* on the US east coast (*DeRivera et al., 2005*).

Benthic species in rocky intertidal communities are commonly prey for birds (*Edwards, Conover & Sutter, 1982*; *Wootton, 1992*). Previous experiments that have excluded avian predators from rocky intertidal habitat found the absence of bird predation caused a significant increase in the density of intertidal crabs like *Cancer borealis* (*Ellis et al., 2007*). Predation of non-native *H. sanguineus* specifically by avian predators has not been well documented but merits additional investigation. Similarly, the relationship between *H. sanguineus* and coastal mammals has yet to be investigated. Further research should include field studies that specifically measure the impact of non-fish predators and the variation in predation risk and refuge throughout the tidal cycle, which could have an effect on the degree of enemy release experienced by non-native *H. sanguineus*.

In the field experiments conducted here, the probability of predation was similar during daytime and nighttime. This is counter to the hypothesis that predation risk would be constrained to the daytime period of the diel cycle, because of the propensity of high tide predators to restrict their activity to daytime. For example, *T. onitis* cease feeding at night (*Dew, 1976*; *Deacutis, 1982*).

Predation at nighttime may have been enhanced because of the proximity of the field experiment to a lighted dock. The dock could have encouraged aggregation of *T. onitis* and *T. adspersus* which are known to affiliate with structures like dock pilings (*Olla, Bejda & Martin, 1975*). Illumination from lights on the dock at night could improve the predators' ability to see prey and forage throughout nighttime, particularly on a dark night, like under cloudy skies or during a new or crescent moon. Nevertheless, man-made structures like the dock described here are increasingly common in coastal habitats (*Ruiz et al., 2000*), and such human influence does affect predation efficiency and prey choice in nearshore ecosystems (*Montalvo, 2020*).

In the laboratory study, High Refuge Habitat significantly reduced the proportion of crabs eaten by *T. onitis*. In addition, there was no difference in the proportion of crabs eaten when provided Low Refuge Habitat and when provided No Refuge. *H. sanguineus* is most abundant in mid and lower intertidal zone where there is high structural complexity (*Ledesma & O'Connor, 2001*; *Brousseau et al., 2002*; *Gilman & Grace, 2009*; *Epifanio, 2013*). A limitation of this study is an under-representation of the true complexity of the rocky intertidal zone, which is more often composed of multiple layers of rocks and cobble that could provide more spatial refuge to further minimize the risk from predators (*Lohrer et al., 2000*). Nevertheless, evidence presented here demonstrates that *T. onitis* will readily consume adult (sexually mature) *H. sanguineus,* and that the availability of structural refuge modifies the risk of predation.

Given the high density of non-native *H. sanguineus,* relative to other intertidal crab species (*O'Connor, 2014*; *O'Connor, 2018*), *H. sanguineus* appears particularly adept at avoiding predation. The high abundance of *H. sanguineus* in many coastal communities throughout the invaded range suggests that predators do not strongly impact the population size of this species, perhaps, in part, because of the temporal and spatial refuge within rocky intertidal habitats.

This study shows that adult *H. sanguineus* can be eaten throughout the diurnal and tidal cycles in the rocky intertidal zone, and spatial refuge strongly modifies predation threat by a common fish predator, *T. onitis.* This study suggests non-native *H. sanguineus* predation risk can be modified by habitat refuge. Subsequent research should combine the factors examined here, spatial and temporal refuge, to directly test if predation pressure experienced by *H. sanguineus* populations is different among coastal habitats with varying structural complexity. Healthy populations of fish could lead to a decrease of *H. sanguineus* abundance.

## ACKNOWLEDGEMENTS

We thank F. Kennedy, E. Hobbs, D. Lavoie, A. Marcelino, K. Oliveira, T. Rajaniemi, Z. Towne and staff of the Massachusetts Department of Marine Fisheries for their assistance with various components of this project.

### Funding

The funding was provided by the Biology Department at the University of Massachusetts at Dartmouth. The funders had no role in study design, data collection and analysis, decision to publish, or preparation of the manuscript.

### Grant Disclosures

The following grant information was disclosed by the authors:
The Biology Department at the University of Massachusetts at Dartmouth.

### Competing Interests

The authors declare there are no competing interests.

### Author Contributions

- Renee C. Montanaro conceived and designed the experiments, performed the experiments, analyzed the data, prepared figures and/or tables, authored or reviewed drafts of the article, and approved the final draft.
- Nancy J. O'Connor conceived and designed the experiments, authored or reviewed drafts of the article, and approved the final draft.

### Animal Ethics

The following information was supplied relating to ethical approvals (i.e., approving body and any reference numbers):

Fish were maintained and housed under UMass Dartmouth Institutional Animal Care and Use Committee protocol # 21-02 approved July 19, 2021.

### Data Availability

The raw data for field and laboratory experiments and R code are available in the Supplemental Files.

### Supplemental Information

Supplemental information for this article can be found online at http://dx.doi.org/10.7717/peerj.16852#supplemental-information.

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
