# Peer review of "Temporal and spatial refugia modify predation risk for non-native crabs in rocky intertidal habitats"

_PeerJ, doi:10.7717/peerj.16852_

## Round 0.1 · original submission · Minor Revisions

I apologize for the delay in completing my review. I received the reviews for your manuscript just after I had started reviewing another manuscript. That manuscript turned out to require a great deal of attention, causing a substantial delay.

Overview
This manuscript presents two experiments assessing predation risk in an invasive crab species. A field study examined the removal (presumed predation) of crabs tethered in the intertidal zone during high and low tide in the day and night. Loss rates during high tide and were much higher than during low tide. For the same tide level, loss rates were similar in the day and night. The authors suggest that fish predators, especially tautog, are probably responsible for most of the predation, but the presence of nocturnal predation suggests another fish or larger crabs may be involved. The laboratory study showed that predation by tautog was greatly reduced in the presence of a refuge for the crabs.

Both reviewers consider the experimental design, analysis and presentation to be sound, needing only a relatively small number of modifications to be publishable. I agree with this assessment, but have some additional concerns not raised by the reviewers. My comments below may be taken as a third review: make appropriate changes if you consider them valid or provide a detailed explanation if you disagree. I will also provide an annotated pdf with minor problems highlighted and suggested changes in inserted comments. Your rebuttal document does not need to include these small comments unless you disagree.

General issues
Length of the Introduction: I agree with the reviewer that the Introduction is rather long. The problem is the somewhat rambling account that serves to introduce potential fish predators and documents tautog as a prime candidate as well as appropriate subject for the experiment. Some shortening can be accomplished by stating your focus on adult stage crabs and then eliminating references to consumption of larval stages. The rest needs to be replaced by a more tightly focused single paragraph.

Species names. At first mention in both Abstract and main body of the article, you should provide scientific names as well as common names if you intend to use them later for all species referred to. After that, use one or the other consistently, but not both. I found that you shifted between the two throughout the manuscript in referring to both the crab and the tautog. If you use the scientific name, also be consistent as to whether you spell out or abbreviate the genus. (Abbreviation would be sufficient.)

Specific comments
Description of the field experiment: When I first read the manuscript, I assumed that the low tide treatment left the crabs out of water. However, given the tide lower intertidal location of the treatments, it seems more likely that the crabs were covered by water during much of the treatment period. Because other readers might make a similar error, I suggest being explicit about the range of water depths over the crabs at both high and low tides and the duration of air exposure during low tide treatments. Similarly, you could add information about how much after sunset the night treatments began. Right at sunset, could there be enough light that diurnal predators would still be active?

L96-103. Since you did not examine predation on megalopae, reduce this to a single sentence.

L128. It is hard to imagine killifish eating adult shore crabs. Is this another reference to megalopae? Including species that eat larvae adds unneeded information and confusion for readers.

L140ff. Consider adding a line indicating that you investigated the temporal refuge by explicitly removing the spatial refuge and that you investigated the spatial refuge by eliminating the temporal refuge. In the Discussion, you might come back to this point to recognize the field implications of the somewhat simplified environments in which the experiments that were carried out.

L147. A reviewer asked what your hypotheses were regarding the outcomes. If you had explicit hypotheses, it would be appropriate to state them here. If you did not have hypotheses but designed your study as an exploration without hypotheses, that is perfectly acceptable, and you and it would not be strong science if you were to generate post-hoc hypotheses.

L167. Be more explicit about what information can be found in Table 1 or remove it and summarize in the text as suggested by the reviewer.

L202. Water depth is potentially important.

L229. How many tanks were used for the experiments? Was the same tank used for the same treatment through the nine trials?

L262-264. I don’t see how the chi square and df can be the same in both comparisons but the p-value very different. (Perhaps I am not understanding something about the test.) I don’t think you need Table 2. Just provide the results in the text, including test statistic, df and p-value (most of it is there already, overlapping with Table 2).

L267ff. Tables 3 and 4 can be removed and any information not provided in Fig. 3 can be added to the text. Also, it is not necessary to add means, SD, and medians in the text because all the needed information is included in Fig. 3.

Figures 1 and 2. I have provided some additional suggestions for the captions of these figures on the pdf.

L285-370. It is appropriate, of course, to consider what predators that may have consumed crabs in field experiments. However, I find this section far too long, speculative and somewhat repetitive. See if you can use a tighter outline to greatly reduce the length and repetition, including with the Introduction. Perhaps four short paragraphs outlining likely or possible predators at daytime high tides, nighttime high tides, daytime low tides and nighttime low tides. The presence of a lighted dock is relevant as part of the picture of the strength of the evidence. Consider also the possibility of overlapping conditions between treatments: presumably the water was fairly high during part of the low tide treatment allowing some access by predators that require deeper water. Was the night treatment fully dark and the day treatment fully light?

L288. It is inappropriate to introduce new results (seining, Table 5) in the Discussion. Two seine hauls provide very weak evidence because of the ease with which they could be missed. You came close to capturing no tautog or cunners! I suggest either removing this entirely or reducing it to a single sentence of text with no table.

L296. I disagree with Reviewer 1 about providing micrographs of the cut end. For it to make a strong difference to the interpretation of predator, you would need to have known examples and categorize all the cut ends.

L323. I presume that you are not implying the blue crabs forage out of water (low tide) but are more likely to continue foraging in shallow water. Because you did not indicate the water depths during the low tide treatment, I initially assumed that the crabs were air exposed during this period.

L340. Note that the term ‘shorebirds’ is often applied to the waders, not gulls, cormorants or ducks. Having asked you to shorten this section, the following comments are not likely relevant except for your general interest. Here on the west coast (I live in Victoria, BC, Canada), glaucous-winged gulls are frequently observed capturing crabs by surface dipping while standing or floating in shallow water. The crabs they capture are possibly larger than Hemigrapsus. According to the online reference, Birds of the World, crabs also form part of the diet of herring and great black-backed gulls found on the east coast, so I don’t think you should dismiss gulls too readily. A brief search turned up YouTube videos of herring gulls catching and eating crabs. On the other hand, Reviewer 1 with experience with this species notes that he has never seen gulls preying on them over a long period of observation. During the summer, our glaucous-winged gulls are on their breeding territories and we don’t see them as much on our shores that are distant from the colonies. This may be relevant, if there were not many foraging gulls present, but not if gulls were present and foraging but not on crabs. Seasonal variation might be of interest for future work and experiments to see if gulls would feed on Hemigrapsus if they are available.

L400-409. It is a good idea to have a short paragraph to conclude the Discussion. Suggestions for future research could be part of this. However, I find this a bit long, overlapping previous points, and rather speculative given your simple experiments. It’s a big leap from a sloping shelter in an aquarium to the diversity of refuges in the field. Since you prevented crabs in the field from using refuges (other than burying in the sand?), perhaps thinking about field experiments to look at the interacting effects of refuge characteristics and water depth would be more relevant. I don’t intend that you present this idea, but hoping that you will think about adding something closer to the next step in this research, especially if you have original insights into approaches from your study that might not occur to others.

L410. Check where Instructions to Authors says to list the permits.

References
The references are generally well done but need some re-checking. Some are incomplete and there are a few missing italics for species names and wayward capital letters. It seems that Collette and Klein-MacPhee is an edited volume, so I would suppose that you need authors of the individual chapters as well as the volume editors in a complete reference.

Reviewer 1 ·

Basic reporting

I find the manuscript to be clear, and professionally presented. Much attention was paid to the literature in support of the rationale and experimental approaches. Results area relevant to hypotheses.

Experimental design

I find the work well done and the reporting detailed. I have only the following comments:
Low Refuge Habitat looks little different the No Refuge control. Figure 3 bears this out.
line 296 provide micrographs of cut end of monofilament to distinguish between abrasion and cut?
line 343 never in 20 years have I seen gulls successfully capturing Hemi

Validity of the findings

Conclusions are well stated, as far as the data allows (i.e., limited to speculation past what the experimental work demonstrated).
line 377 Lohrer et al. (2000) in JEMBE did the original structural complexity work.
Fig. 2 I am surprised that so many actually survived tethering at low tide. Argues against gulls as significant predators, seems to me.
delete Table 1 and integrating the important descriptive information into the Materials and Methods text.

Additional comments

line 314 future: would be nice to see the tethering across the intertidal zone

·

Basic reporting

This manuscript investigates the predation intensity on the invasive Asian shore crab (Hemigrapsus sanguineus), which is now one of the most common intertidal crab species on US Northeast Atlantic shores. The authors performed one experiment in the field (using tethering) and another in the lab to assess the survival of crabs considering a range of factors, such as the tide regime and period of the day (for the field experiment), and the complexity of refugia provided to crabs (for the lab experiment). Below, I detail important issues demanding consideration within the “Basic Reporting” section:

1 - the manuscript is well-written and the English is clear. Similarly, the referenced literature is compatible in all places, covering important papers in the area.

2 - I noticed that the authors used two different ways to refer to the Asian shore crab in the text, either as a "non-native" or as an "invasive" species. It is probably better to choose one of them since the two words may refer to different moments of the invasion process of a species.

3 - The Introduction section is very long and composed of many short paragraphs. I think you can either join subsequent paragraphs that are connected in some way or remove some of the information, placing it in other sections. For example, the text between lines 104 and 126 (where you talk about the possible fish predators of Asian shore crabs) is not fundamental for this part of the story and can be reduced a lot or even placed in the methods section when you describe your lab experiment and why you used that fish species as crab predator.

4 - There is no reference to your hypothesis and predictions in the last paragraph of the Introduction. What do you expect to find in your experiments and why?

Experimental design

The knowledge gap that this research tries to cover is clear and well-explained. Also, the protocol of the two experiments is correctly described and understandable, but there are some concerns that I detail below:

1 - Why is the number of crabs used in the field experiment (34) so low compared to the sample size used in the lab experiment (720)?

2 - I did not understand how many replicates you had for the tethering field experiment. You say that you used 34 crabs, but this number is for all trials you ran (all 4 combinations) or for each one of the combinations (i.e. 34 at low tide during the day, 34 at low tide during the night, and so on). Be clear about the number of replicas for this experiment.

3 - In the statistical analysis of the experiments you explain that you first used full models testing the effects of other factors but because these were not significant you decided to keep only the reduced models. I think you could provide these initial models with the associated p-values in the R script that you added as supplementary material so we can understand your decision to keep with the simplified models.

Validity of the findings

1 - Thanks for sharing the raw data. They are complete and understandable.

2 - Tables 2 and 3 show the results of very simple analyses. Based on that, I think you could add the results of the stats directly to the text when you explain your findings. Similarly, Table 4 shows only the post hoc comparisons, and providing this as a table in a manuscript is not common since you already have such comparisons shown in Figure 3 (the letters above the boxplots).

3 - Figure 3 would benefit from a better aesthetic visualization. You can add the raw values as jitter points behind or ahead of the boxplots so we can observe the natural variation of the data.

4 - The results are well discussed but as occurred in the Introduction I also thought that the Discussion section is too long and lots of information that is provided is not essential and could be removed.

5 - I lacked at the end of the Discussion something about how your results could be used to address all the issues related to the invasion of the Asian shore crabs on the US coast. Do you think some of your findings could be applied to control the size of the crab population at the invaded places?

Additional comments

I added the pdf with some minor comments that were not addressed in the sections above.

---

## Round 0.2 · accepted · Accept

The article is now suitable for publication. I appreciate the authors' thorough consideration of the comments of reviewers and editor and the detailed rebuttal.